# Ras and Wnt Interaction Contribute in Prostate Cancer Bone Metastasis

**DOI:** 10.3390/molecules25102380

**Published:** 2020-05-20

**Authors:** Shian-Ren Lin, Ntlotlang Mokgautsi, Yen-Nien Liu

**Affiliations:** 1Graduate Institute of Cancer Biology and Drug Discovery, Collage of Medical Science and Technology, Taipei Medical University, Taipei 11024, Taiwan; d9813003@gms.ndhu.edu.tw; 2Ph.D. Program for Cancer Molecular Biology and Drug Discovery, College of Medical Science and Technology, Taipei Medical University, Taipei 11024, Taiwan; d621108006@tmu.edu.tw

**Keywords:** Wnt, Ras, prostate cancer, bone metastasis, cross reaction

## Abstract

Prostate cancer (PCa) is one of the most prevalent and malignant cancer types in men, which causes more than three-hundred thousand cancer death each year. At late stage of PCa progression, bone marrow is the most often metastatic site that constitutes almost 70% of metastatic cases of the PCa population. However, the characteristic for the osteo-philic property of PCa is still puzzling. Recent studies reported that the Wnt and Ras signaling pathways are pivotal in bone metastasis and that take parts in different cytological changes, but their crosstalk is not well studied. In this review, we focused on interactions between the Wnt and Ras signaling pathways during each stage of bone metastasis and present the fate of those interactions. This review contributes insights that can guide other researchers by unveiling more details with regard to bone metastasis and might also help in finding potential therapeutic regimens for preventing PCa bone metastasis.

## 1. Introduction: Metastasis of Prostate Cancer (PCa)

Bone marrow is the most frequent metastatic site of PCa, which include 72.8% of all metastatic cases, at over three times higher than the second most frequent site [1]. Knowing the underlying mechanism of osteophilic nature in PCa metastasis is urgent for monitoring and preventing PCa bone metastasis. Interestingly, PCa metastasis could be divided into two types: osteoblastic and osteoclastic, and that is apparently triggered by different mechanism [2]. Which molecules or cells are key factors in choosing the bone metastasis type? These two questions have attracted the attention of researchers in cancer biology, in the hope that they can be applied to an early diagnosis of cancer metastasis.

Wnt signaling, either canonical or non-canonical, leads to PCa development, especially in castration-resistance PCa (CRPC) and metastasis [3]. The Ras superfamily includes Ras, Rho, and Ral, and it is known to be involved in almost all fundamental cytological activities [4]. Cumulative studies have unveiled that crosstalk between Wnt and Ras signaling is highly associated with the tumorigenesis and progression of various cancers, *e.g.*, colorectal cancer and mucinous ovarian cancer [5,6,7]. In this review, we collected and discuss recent findings on the role of Ras/Wnt interactions in bone metastasis (BM) of PCa and try to emphasize their regulatory dynamics in different metastatic stages.

## 2. Mechanism of BM in PCa

The overall progression of PCa BM can be characterized into three main steps: cancer cell invasion and circulation, interaction with osteocytes, and homing toward bones [8]. The following section describes a recently published mechanism of PCa BM that is based on the above three steps.

### 2.1. Epithelial-Mesenchymal Transition (EMT) and Tumor Invasion

#### 2.1.1. TGF-β/MAPK-Mediated EMT

During metastasis, PCa cells transform into mesenchymal-like cells, which helps cells to invade through the basal lamina and survive in the blood circulation without cell-cell interactions [9]. The EMT in PCa is activated by four mediators of snail, slug, twist, and Zinc finger E-box-binding homeobox 1 (ZEB1), while four signaling pathways play dominant roles here: the transforming growth factor (TGF) receptor (TGFR)/mothers against decapentaplegic homolog (Smads)/mitogen-activated protein kinase (MAPK) signaling cascade, nuclear factor (NF)-κB signaling, and the receptor tyrosine kinase (RTK) cascade [9]. Wa et al. reported that MAPK can also promote the EMT through inhibiting Rb phosphorylation, which is one of the main suppressors in cell cycle regulation [10]. Other signaling factors regulate the TGF-β-MAPK axis in PCa BM. Lue et al. described how MAPK is upregulated by heparin-binding epidermal growth factor (HB-EGF)-mediated signaling, in which the HB-EGF is cut from the PCa surface by the zinc transporter, ZIP6 (LIV-1), which stimulates matrix metalloproteinase (MMP)-2/9 upregulation in other PCa cells [11]. Interferon (INF)-induced transmembrane protein 3 (IFITM3), a highly expressed transmembrane protein in cancer cells, is believed to be associated with metastasis in various cancers, *e.g.,* acute myeloid leukemia, hepatomas, and gliomas [12,13,14,15]. In PCa, IFITM3 promotes fibroblast growth factor (FGF) expression and promotes TGF-β production, which elicits the EMT in neighboring cells in a paracrine manner [16]. On the contrary, the TGF-β-mediated EMT can be retarded via microRNA (miR) regulation. miR-33a-5p reduces TGFR 1 expression, which affects its offset by increasing the ZEB1 copy number [17]. Moreover, the TGFR and Smad2/4 are suppressed by miR-505-3p and miR-19a-3p [10,18]. Those studies clearly depicted a regulatory network in TGF-β-mediated BM in PCa cells.

#### 2.1.2. NF-κB Activation after Androgen Receptor (AR) Signaling Deprivation

NF-κB signaling pushes cancer metastasis in multiple directions, such as stimulating MMP expressions and regulating cell adhesion molecules, according to previous studies [19]. The tumor necrosis factor (TNF)-α receptor (TNFR) promotes inhibitor of NF-κB (IκB) kinase (IKK) activity, which blocks the binding of IκB to NF-κB and releasing the active form of NF-κB [20]. Active NF-κB ultimately triggers hypoxia-inducible factor (HIF)-1α expression and subsequently induces the EMT [21]. In addition to TNFR signaling, NF-κB can also be activated by TNF-related weak inducer of apoptosis (TWEAK)/TNFR superfamily member 12A (TNFRSF12A, also known as Fn14)-mediated IKK-β activation and downregulation of miR-210-3p-triggered suppressor of cytokine signaling 1 (SOCS1) and TNFAIP3-interacting protein 1 (TNIP1) [22,23]. Conversely, activated AR and its cofactor FOXA1 inhibits TWEAK/Fn14/IKK-β activation through directly binding to an androgen-binding element in TWEAK and the Fn14 promoter/enhancer in order to reduce TWEAK and Fn14 transcription [22]. After androgen deprivation therapy (ADT), some castration-sensitive PCa cells will transit into CRPC cells, which is the beginning of PCa metastasis [24,25]. Izumi and Mizokami summarized the characteristic of C-C motif ligand 5 (CCL5) in regulating AR expression, in which CCL5 downregulates AR expression [26]. The above studies not only evaluated the second central signaling axis in PCa BM, but also evaluated how CRPC is induced.

#### 2.1.3. Contribution of PI3K/Akt/MAPK Signaling in EMT of PCa

The third signaling pathway that is involved in PCa BM is the phosphoinositide 3-kinase (PI3K)/Akt signaling cascade, which originates from the activation of the epidermal growth factor (EGF) and vascular endothelial growth factor (VEGF). In general, the activation of EGF and VEGF receptors (EGFR and VEGFR) stimulates the Ras/Raf/MAPK kinase (MEK)/MAPK signaling cascade, which is involved in tumor progression or the PI3K/Akt/mammalian target of rapamycin (mTOR) cascade that promotes cell growth and the EMT [27,28]. In PCa, EGF signaling accompanies alterations in miR-96 and miR-30 expression, which act contrary to each other. EGF signaling promotes miR-96 expression, which attends to the degradation of E26 transformation-specific variant 6 (ETV6, also known as TEL, a transcriptional repressor in regulating embryonic and hematopoietic cell proliferation) that blocks the expression of the TWIST1 oncogene [29,30,31,32]. Kao et al. reported that EGF signaling inhibits miR-30 expression, which directly reduces ETS-related gene (ERG) expressions [33]. In addition to EGF signaling, miR-30 can also be reduced by Src/STAT3, which is mediated by the VEGFR/NRP-1/c-Met/Mcl-1 cascade [33,34]. When tracing upstream of VEGF signaling in PCa metastasis, reprogramming of glucose metabolism was identified as a critical step for the EMT [35]. The core regulator of glucose metabolism, AMP-activated protein kinase (AMPK), triggers cell migration-inducing protein (CEMIP) overexpression through the AMPK/glycogen synthase kinase 3β (GSK3β)/β-catenin cascade for which CEMIP mediates VEGF and MMP-2 upregulation and subsequently results in anoikis resistance [36]. In addition to AMPK, VEGF expression can also be modulated by HIF-1α. The RTK signaling cascade promotes mTOR phosphorylation, which elevates HIF-1α expression [37]. Furthermore, HIF-1α triggers pyruvate kinase M2 (PKM2) as a transcription factor that stimulates neuroendocrine markers, like oct4 and VEGF [38,39]. The EMT can be activated by PI3K/Akt- and MAPK-mediated mTOR activation, which promotes EMT and metastasis through the phosphorylation of eukaryotic translation initiation factor 4E-binding protein 1 (EIF4EBP1) [40,41,42]. Bi et al. and Tang et al. demonstrated that miR-153 and miR-133a-3p are involved in PCa BM, in which miR-153 exacerbates the EMT through inhibiting phosphatase and tensin homolog (PTEN), and miR-133a-3p acts inversely through reducing growth factor receptor expressions [41,43]. Those studies provided further insights into RTK signaling in the EMT, and not just in maintaining cell survival [44].

#### 2.1.4. Other Minor EMT contributors

Other minor mediators that are discovered to be associated with PCa BM include KDM8, miR-145, and CCCTC-binding factor (CTCF). In the previous paragraph, we discussed the inhibitory characteristics of the AR in PCa metastasis. However, KDM8, an AR transcriptional enhancer, which can accompany with AR binding to the enhancer and promoter region of ATPase family AAA domain-containing protein 2 (ANCCA) and enhancer of zeste homolog 2 (EZH2) gene, results in PCa metastasis, neuroendocrine differentiation, and growth promotion [45]. miR-145 is a tumor-suppressing microRNA in various cancers, including PCa, head and neck squamous cell carcinoma, and breast cancer [46]. Reports from Guo *et al.*, Ren *et al.*, and Huang et al. showed that miR-145 is activated by p53 and it inhibits mesenchymal and cancer stem cell markers, like fibronectin, vimentin, cluster of differentiation 44 (CD44), and c-Myc via inhibiting two mediators: human enhancer of filamentation 1 (hEF1) and Krüppel-like factor 4 (KLF4) [47,48,49]. However, the AR and KLF4 form reciprocal feedback for promoting expression and that also results in the inhibition of PCa proliferation and metastasis [50]. KLF4 is a crucial regulator of normal cell proliferation and it can be a tumor suppressor or an oncogene, depending on the tissue, tumor type, or cancer stage [51,52,53]. miR-1 acts as a tumor suppressor that is involved in regulating BM, and miR-1 is tightly regulated by the AR [54]. Siu et al. demonstrated that KLF4 functions as a transcription factor to activate the AR-miR-1 signaling pathway to constrain the tumor-suppressive role of miR-1 [50]. CTCF, a downstream transcription activator of Smad and Myc signaling, contributes to slowing down metastasis in PCa through directly interacting with the miR-127-3p promoter, in which the downregulation of miR-127-3p accompanies proteasome subunit beta type-5 (PSMB5) elevation and subsequently leads to the activation of BM [55,56,57]. All of the above research draws a large picture of the regulatory mechanism of BM in PCa cells, and Figure 1 summarizes these signaling cascades. In the next section, we discuss how PCa cells interact with bone stromal cells and the function of PCa/osteocytes in BM.

### 2.2. PCa/Osteocyte Interactions

#### 2.2.1. Surface Modulation of PCa during Circulation

Why do circulating PCa cells migrate to bone? This question can be answered by the interactions among osteocytes, bone marrow stromal cells (BMSCs), and PCa cells. In the circulation, PCa cells need to against attack by immunocytes, especially natural killer (NK) cells [58]. Accordingly, tumor cells (not just PCa cells) express several surface markers or immunocyte regulatory factors to escape NK cells. With PCa cells, the stem cell marker, nanoG, was proven to help PCa cells to escape destruction by NK cells through downregulating intercellular adhesion molecule 1 (iCAMP1) [59]. Additionally, nanoG also triggers various cancer stem cell markers and chemokine receptors, such as C-X-C motif receptor 4 (CXCR4), CD133, aldehyde dehydrogenase 1 (ALDH1), and insulin-like growth factor (IGF)-binding protein 1 (IGFBP1) [60]. Meanwhile, interferon (INF) regulatory factor 7 (IRF7, generally known as main regulator of interferon I response in innate immune response), is secreted by PCa cells, which can counteract NK cell activation through modulating IFN-β-mediated signaling [61,62].

#### 2.2.2. Circulated PCa Interacts with Osteocytes

PCa cells are attracted to chemokines that are released from osteocytes and BMSCs and migrate toward the bone marrow [63]. In the meantime, PCa cells feedback protein signals to osteocytes that augments growth factors secretion from osteocytes and further trigger osteocyte differentiating into osteoblast (release growth factors and build a calcium-rich environment) or osteoclast (produce a niche for PCa homing) [64,65,66]. As PCa cells sense CCL5 and other chemokines, such as C-X-C motif chemokine 12 (CXCL12) produced by BMSCs and osteocytes, PCa cells stimulate osteocytes to secrete growth-derived factors 10 and 15 (GDF10 and GDF15) that can push PCa cells secreting early growth response 1 (EGR1) and parathyroid hormone-related protein (PTHrP), two important factors for osteoclastogenesis [2,67,68]. Additionally, the transcription factor, Runt-related transcription factor 2 (RUNX2, an essential factor in osteogenesis which is hyperactivated in PCa cells) is phosphorylated by the integrin a_v_β_3_/Smad5 cascade, CD44, and integrin a_v_β_3_/Src/rac1 signaling and activates CXCR7 and Akt, which upregulate NF-κB resulting in receptor activator of NF-κB ligand (RANKL) overexpression, which can induce osteoclastogenesis [69,70,71,72,73,74]. Different from osteoclastogenesis, the role of osteoblastogenesis is to help circulating PCa cells landing in the bone [75]. Therefore, interactions between circulating PCa cells and osteoblasts can cause cell adhesion molecule expressions [65]. These include kallikrein-related peptidase 4 (KLK4), which is the downstream signaling transducer of protease-activated receptor 1 (PAR1) that can elevate MMP-1 expression leading to the release of thrombospondin 1 (TSP1, an osteoblast inducer) [76,77]. At the same time, osteoblasts secrete vascular cell adhesion molecule 1 (VCAM1) and collagen, which help PCa cells adhering to osteoblasts with sonic hedgehog (Shh)/Gli/PTCH1 signaling activation, and those successfully landing directly increase alkaline phosphatase 2 (AKP2) expression and ultimately trigger bone rearrangement [78,79,80]. Those studies provided clear insights into PCa/osteocyte interactions and answered the question about the selective metastasis of PCa cells.

### 2.3. PCa Homing

As PCa cells land in bone marrow, a series of tissue changes occurs, including angiogenesis, the escape of PCa cells from dormancy, and niche formation [81]. In niche formation, BMSCs, osteoblasts, osteoclasts, hematopoietic stem cells (HSCs), monocytes, and pericytes secrete growth factors, and calcium around PCa cells that help PCa cells remain dormant for self-renewal [65,82,83]. Growth factors that are enriched in the tumor microenvironment (TME) include TGF-β, Ca^2+^, hematopoietin, bone morphogenic proteins (BMPs), CXCL12, annexin A2 (ANXA2), and insulin-like growth factor (IGF) [2,82,84,85], among which TGF-β and BMPs play central roles in holding PCa cells in dormancy via the tyrosine-protein kinase receptor, UFO (Axl)/growth arrest specific 6 (Gas6) axis [86,87]. Localized PCa cells are reactivated when the growth suppressors (TGF-β and BMPs) are removed or growth conditions occur. The trigger for reactivation from dormancy is not exactly known; however, several hypotheses and phenomena were proposed. Giancotti provided a hypothesis in his review that presumes that the epithelial-mesenchymal transition (EMT) might be a possibility for reactivation from dormancy [88]. A review by Byrne et al. summarizes that castration causes bone loss and increased metastasis via hypoxia induction and vascular remodeling [89]. Furthermore, Miftakhova et al. reported that cyclin A1 coupled with aromatase (CYP19A1) enables PCa metastatic growth in bone marrow through manipulating androgen concentrations in the TME [90]. Barkan et al. found that dormant PCa cells are induced by a collagen-I-enriched condition, which implies the importance of integrin/extracellular matrix (ECM) interactions in reactivating dormant PCa cells [91]. Finally, Dai et al. reported that PCa cells excrete PKM2 toward BMSCs and adipocytes in bone marrow through exosomes to promote HIF-1α upregulation in BMSCs [92]. The upregulation of HIF-1α increases CXCL12 secretion and induces PCa cell proliferation and the Warburg effect [92]. These hypotheses and phenomena shed some light on understanding the detailed mechanisms of PCa BM.

In the above discussion, we collected and summarized recent knowledge of the mechanisms regulating PCa BM. In the following section, we discuss the impacts of the ras/MEK axis and Wnt/β-catenin signaling pathway in PCa BM and their crosstalk.

## 3. Impact of Wnt and Ras Signaling in PCa BM

### 3.1. Ras Signaling

Ras signaling is known to be involved in the proliferation and metastasis of various cancers, including breast cancer, colorectal cancer, and lung cancer [93]. In breast cancer, Ras signaling mainly promotes metastasis through activating MEPK, STAT3, and PI3K, and then triggers the NF-κB-mediated EMT [94,95]. In PCa, Ras forms a huge signaling network for regulating BM for which the Ras/Raf/MEK/extracellular signal-regulated kinase (ERK)/Elk-1 signaling cascade is the central backbone [96]. Elk-1 is a transcription factor that directs gene expression of c-Myc and its co-transcription factor, Myc-associated zinc-finger (MAZ) [97,98]. MAZ upregulates Ras expression and then activates NF-κB through the RalGEF/Ral/NF-κB cascade and finally activates the EMT [97,99]. In addition, NF-κB is involved in the heterotrimer G-protein signaling network, which mediates cell adhesion signals through guanine nucleotide-binding protein subunit α-15 (GNA15), and Rho, which promotes CXCL5 expression [100,101]. Another important branch of the Ras-signaling network is the Ras/PI3K/Akt signaling cascade, which transduces signals from growth factor receptors and promotes PCa metastasis, as described in the previous section [102]. MAPK members, ERK1/2, are key downstream regulators of the Ras-signaling network [103]. In addition to Ras signaling, ERK1/2 also receive signals from sprout homolog 2 (SPRY2)/protein phosphatase 2A (PP2A) and Ras-related protein 2B (Rap2B)/focal adhesion kinase (FAK) and activates Runt-related transcription factor 2 (RUNX2) or ETS/AP1 which are essential for PCa/osteocyte interactions [72,104,105,106,107]. Several activators were identified upstream of Ras. In addition to canonical signals from the EGF receptor (EGFR) through SHC/GRB2/SOS phosphorylation, other upstream activators contain ETV4 [108], Ras-related protein 25 (Rab25) through Src [109], RasGRP3, which naturally activates Ras via helping Ras-GDP release and that could be promoted by endothelin A receptor (ET_A_R) [110,111], E-selectin ligand 1 (ESL-1), which controls the rolling capacity of circulating PCa cells [112], miR-let-7c, which possibly promotes c-Myc [113,114,115], miR-407, which is secreted by cancer-associated fibroblasts that block Ras suppressor 1 (RSU-1) [116], Ras and a-factor-converting enzyme 1 (Rce1) by acting on prenylation that works on Ras anchoring [117,118], EZH2 by blocking disabled homolog 2-interacting protein (DAB2IP), a GTPase-activating protein that converts Ras-GTP to Ras-GDP, and stabilizes HIF-1α for the EMT [119,120,121], and the oxytocin receptor (OXTR) by coupling with the EGFR [122]. In contrast, to retard Ras signaling during PCa metastasis, only miR-145 [123] and phosphoprotein associated with glycosphingolipid-enriched microdomains 1 (PAG)/RasGAP [124] were identified. Summarizing those studies, we found a role of the Ras signaling network in PCa metastasis in activating NF-κB and MAPK members. In the next section, we continue discussing recent findings of Wnt signaling in BM of PCa.

### 3.2. Wnt/β-catenin

The Wnt signaling pathway is essential for prostate carcinogenesis, which is involved in the formation of the castration-resistant phenotype [3]. Recent studies further revealed that Wnt signaling also takes part in BM, especially in PCa/osteocyte interactions and the homing stage [3]. Wnt signaling can be categorized into canonical and non-canonical pathways, in which the canonical pathway passes signals through LDL receptor-related protein 6 (LRP6), Frizzled (FZD), β-catenin, and TCF/LEF transcription factors, while the non-canonical pathway uses phospholipase C (PKC), DAAM1, c-Jun N-terminal kinase (JNK), and NFAT as messengers instead of β-catenin [125]. In PCa metastasis, canonical and non-canonical types of Wnt pathways have different roles. A series of studies on the regulatory role of dickkopfs-1 (DKK-1, a Wnt signaling inhibitor) in PCa metastasis reported that metastatic PCa cells secrete PTHrP toward osteocytes, which could decrease DKK-1 expression in PCa cells in a paracrine manner [126]. DKK-1 can block the intratumoral canonical Wnt/TCF signaling pathway through the Kremen receptor and decrease osteoblastogenesis [127,128]. Concurrently, DKK-1 further promotes tumor growth via non-canonical Wnt/JNK signaling [128]. Pontillo et al. reported that Wnt5a (a member of the Wnt family) promotes metastatic PCa cell dormancy via the non-canonical Wnt5a/ROR2/SIAH2 cascade [129]. Rabbani et al. extended the characterization of Wnt signaling to PCa metastasis, in which Wnt promotes PCa cells to undergo the EMT via an LRP5-mediated canonical pathway [130]. Those results indicate different characteristics of the canonical/non-canonical pathways of Wnt in PCa metastasis. Notably, the canonical and non-canonical pathways can cooperate for the same consequences by specific regulators. Vela et al. reported that the activation of Wnt5a promotes PCa metastasis through canonical and non-canonical activation of PITX2 [131].

In PCa/osteocyte interactions, Wnt signaling plays a role in transmitting signals between PCa cells and osteocytes. Noggin is a signaling protein that is involved in bone and neuron development [132]. PCa cells regulate the osteoclastogenesis/osteoblastogenesis balance by secreting noggin toward osteocytes, and this results in the Wnt inhibition of osteocytes [133]. Moreover, WNT1-inducible signaling pathway protein 1 (WISP1) is a downstream effector of Wnt signaling, which promotes osteoblastogenesis through upregulating osteocytic BMP2/4 and osteopontin expressions [80,134]. Sclerostin (SOST) is an osteocyte-derived glycoprotein that is known as a Wnt inhibitor and it is highly involved in osteoporosis [135]. In PCa metastasis, PCa cells inhibit SOST secretion and therefore promote Wnt signaling by osteoblasts. Increased osteoblastic Wnt signaling can activate long non-coding RNA MALAT1 expression in PCa cells through an unknown mechanism [136]. Beyond PCa/osteocyte interactions, Wnt signaling in metastasis can be modulated by p53, SETDB1, ERG, and TBX2. p53, a prominent tumor-suppressor gene that not only acts as a cell cycle keeper and apoptotic inducer in cell growth, but also reduces PCa metastasis through downregulating the expression of the Wnt receptor, FZD8 [137,138]. Transmembrane protease serine 2:v-ets erythroblastosis virus E26 oncogene homolog (TMPRSS2:ERG) gene fusion is common in PCa and it is critical for PCa development [139,140]. Chakravarthi et al. demonstrated that the role of TMPRSS2:ERG in PCa development is to elevate Wnt signaling through upregulating FZD4/8 expression [141]. T-box transcription factor 2 (TBX2) is associated with the EMT by modulating epithelial and mesenchymal cell marker expressions via elevating Wnt3a expression [142,143]. Histone-lysine N-methyltransferase (SETDB1) is known as an oncogene that adjusts gene expressions through histone methylation [144,145]. A proteomic study in different stages of PCa cell lines revealed a converse relationship between SETDB1 and Wnt signaling, which might be related to BM of PCa [146]. Those studies emphasized the importance of Wnt signaling in PCa BM, especially in PCa/osteocyte interactions. In the next section, we discuss recent findings about Ras/Wnt crosstalk in PCa metastasis.

## 4. How Ras/Wnt Crosstalk Affect PCa BM

In the above sections, we summarized recent information on the roles of Wnt and Ras signaling in PCa metastasis. This information elucidates that Wnt signaling and Ras signaling might be responsible for different cytological activities, whereas they could synergistically accelerate PCa tumor progression [147]. In fact, Ras and Wnt signaling act reciprocally in PCa/osteocyte interactions and the bone-homing stage, in which Ras signaling promotes osteoclast-specific factor expressions and Wnt signaling helps PCa cells to attach to osteoblasts [148]. Moreover, Browne and colleagues reported that Ras signaling upregulates DKK-1 expression, which inhibits Wnt3a expression and reduces osteoblastogenesis in the osteolytic phenotype of PCa tumors [149]. It is possible that Wnt signaling and Ras signaling cooperate for particular purposes. In research conducted by Chang *et al.*, Wnt signaling helped circular PCa cells attach to osteoblasts via the secretion of WISP-1 in a paracrine manner. WISP-1 activates endothelin-1 (ET-1)/ET-1 receptor signaling, promotes MAPK signaling, and subsequently elevates α_4_β_1_ integrin expression, which is essential for PCa/osteoblast attachment [80].

In maintaining PCa stem cell homeostasis, mixed lineage kinase 3 (MLK-3), a MAPK member, promotes β-catenin’s interacts with KLF4 instead of TCF, and finally reduces EMT initiation [150,151,152]. When in circulation, PCa cells need to overcome nutrient limits, which might cause tumor necrosis. Therefore, PCa cells trigger macropinocytosis for nutrient scavenging and undergo metabolic reprogramming to overwhelm that scenario [36,153]. Macropinocytosis and metabolic reprogramming are Ras-dependent and they are mediated by AMPK activation and PTEN loss [153]. AMPK phosphorylates GSK3β at Ser^9^, which inactivates its activity and results in β-catenin retention [36]. Those reports reveal the diverse roles of Ras signaling and Wnt signaling in PCa metastasis.

Despite the diverse effects on PCa metastasis, Ras signaling and Wnt signaling are co-regulated by several upstream regulators. Rab25, a member of small GTPase that belongs to the Ras superfamily, is known as a direct activator of Ras signaling and Wnt signaling in various cancers, including hepatocellular carcinoma and non-small-cell lung cancer, but not yet in PCa [154,155]. In PCa, the Rab expression level was found to be positively interlocked with malignancy and recurrence after a radical prostatectomy [156]. It is expected that Rab also plays roles in upregulating Ras and Wnt signaling in PCa metastasis. Angiomotin (Amot) is a component of tight junctions that are important for maintaining apical/basal polarity in epithelial cells [157]. In various types of cancer, Amot promotes cell proliferation and alleviates metastasis through elevating MAPK expression and reducing β-catenin [157]. In PCa, Amot was proven to regulate cell growth via the Hippo/YAP signaling pathway, which is downstream of Wnt signaling [158]. Accordingly, Amot might contribute to co-regulating Ras and Wnt signaling in PCa cells. Chen et al. illustrated that miR-34a acts as a tumor suppressor in PCa through downregulating both TCF7 and BIRC5, which are Wnt and Ras signaling effectors, respectively [159], suggesting an interplay between WNT signaling and miR-34a expression in Ras-activated PCa. Other upstream regulators, like sphingosine kinase 1/2 (SK1/2) [160], the TMPRSS2:ERG fusion mutation [161], and ss5TMD4 [162], were demonstrated to modulate both Wnt and Ras signaling. In conclusion, these aforementioned studies emphasized the tight crosstalk between Ras and Wnt signaling in PCa metastasis and point to expected Ras/Wnt co-regulators that have been proven in other cancers. These mechanisms are critical for PCa metastasis, particularly in the osteoblastogenesis/osteoclastogenesis switch and EMT/proliferation switch.

## 5. Conclusions and Future Prospects

In this review, we elucidated the crosstalk between Wnt signaling and Ras signaling in BM of PCa cells, which is summarized in Figure 2. High interactions between Wnt and Ras signaling reveal the therapeutic possibility of PCa BM by modulating Wnt/Ras interactions. Overall, Ras signaling seems on the upstream of Wnt signaling and that guide PCa cell to local proliferation rather than EMT, instead of Wnt. However, during PCa homing, osteoblastogenesis that is triggered by Wnt-mediated signal protein release is essential for homing. Accordingly, it is merely impossible to identify which signaling molecules in Ras and Wnt are the overall key factors of PCa BM. Furthermore, compensation for Wnt and Ras signaling might be considered as one pathway for inhibition or reduction.

Moreover, the above studies all exhibited that the roles of Wnt and Ras signaling in PCa progression are not always concurrent. Sometimes they are antagonistic to each other. BM of PCa occurs through consecutive progression with different signaling modulators in which external growth factors are involved. What is the upstream modulator of Wnt and Ras in PCa progression? Recently, this question was partially answered. The dynamics of Wnt and Ras at each stage of PCa metastasis need to be clarified, and the details within each stage must be elucidated, in order to completely answer the question. Byne et al. have reviewed that castration-mediated osteoporosis might link to PCa BM which bone reformation might increase BM activity [89]. Detail mechanism regarding castration-mediated osteoporosis is not well-known but recent knowledge indicates that Wnt signaling is involved in osteoblast activity instead of osteoclast activity [163]. This information implies that castration might augment bone resorption and that upregulate Wnt signaling in osteocytes and further trigger BM. As more details of Wnt/Ras interactions become known, these developments can be used in order to develop additional preventive and therapeutic targets.

## Figures and Tables

**Figure 1 molecules-25-02380-f001:**
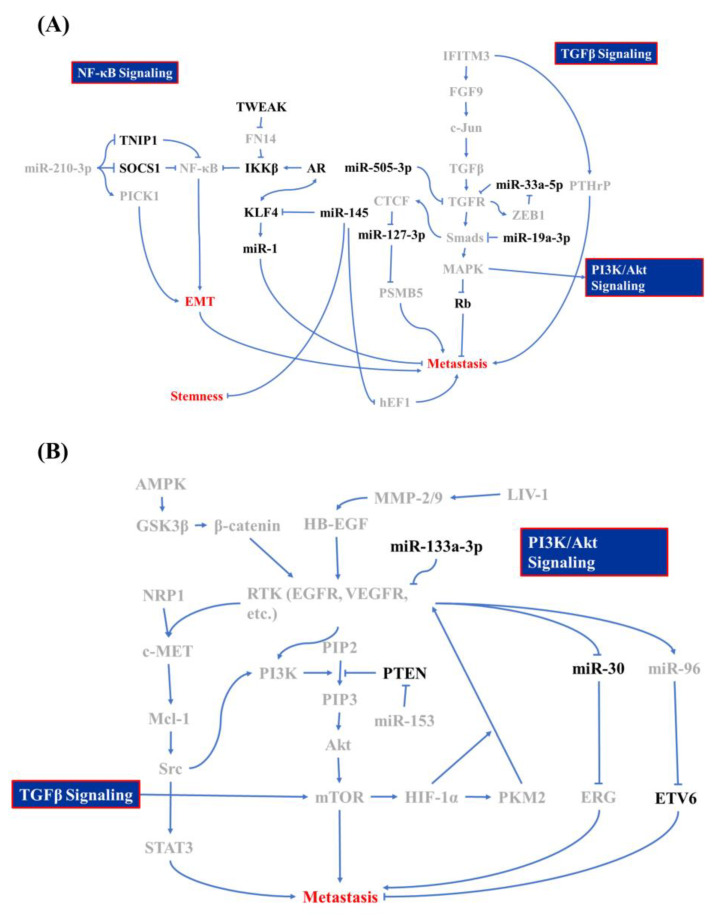
Molecular mechanism of bone metastasis in prostate cancer. Signaling regulators of (**A**) the nuclear factor (NF)-κB, transforming growth factor (TGF)-β, and (**B**) phosphatidylinositol-3-kinase (PI3K)/Akt signaling cascade whose is involved in bone metastasis of prostate cancer and which acts as promoters are labeled in gray and inhibitors are in black.

**Figure 2 molecules-25-02380-f002:**
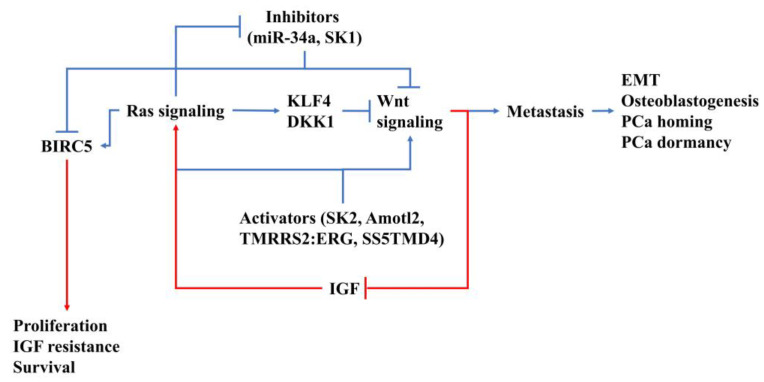
Diagram of Ras/Wnt interactions during prostate cancer (PCa) bone metastasis. The crosstalk from Ras to Wnt is marked in blue and vice versa in red.

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
