# Peer review of "Ras and Wnt Interaction Contribute in Prostate Cancer Bone Metastasis"

_molecules, 2020, doi:10.3390/molecules25102380_

Round 1

Reviewer 1 Report

The manuscript by Lin et al. is a review about the complex interactions between the Wnt and Ras signaling pathways during the progression of bone metastasis derived from prostate cancer. The manuscript is well written in an engaging and lively style. Given the complexity involved, the author has produced many positive and welcome outcomes. The literature review offers a useful overview of current research and policy, and the resulting bibliography provides an especially useful resource for current practitioners.

I have no hesitation in recommending that it be accepted for publication in Molecules after a few typos and other minor details have been attended to.

- Line 25: “…second most frequent site”. Which is the first?

- Lines 26-28: I can not understand this sentence. I mean, it does not make sense here. Please, delete it.

- Line 35: “various cancers”. Please add another example

- Figure1: This figure is extremely useful but, the resolution and the quality are very low. Please increase the quality of this image.

- Line 137: inhibiters? Maybe you can say “inhibitors”?

- Line 148: …..NK cell regulatory factor, viz.,. Is it a typo?

- Line 151-155: This sentence is too large and hard to follow. Please, rephrase

Author Response

The manuscript by Lin et al. is a review about the complex interactions between the Wnt and Ras signaling pathways during the progression of bone metastasis derived from prostate cancer. The manuscript is well written in an engaging and lively style. Given the complexity involved, the author has produced many positive and welcome outcomes. The literature review offers a useful overview of current research and policy, and the resulting bibliography provides an especially useful resource for current practitioners.

I have no hesitation in recommending that it be accepted for publication in Molecules after a few typos and other minor details have been attended to.

 - Line 25: “…second most frequent site”. Which is the first?

A: We thank reviewer’s suggestion. Reviewer might misunderstand the description. Bone metastasis is the most frequent metastatic site and the second site is visceral disease like lung.  

- Lines 26-28: I can not understand this sentence. I mean, it does not make sense here. Please, delete it.

A: We appreciate reviewer’s suggestion. In fact, these two sentences describe the main issue of studying PCa bone metastasis so that we think it need to be kept. We have edited these two sentences which make them more logic from following. Please check them out at Line 25-28.

 - Line 35: “various cancers”. Please add another example

A: We thank reviewer’s suggestion. We have added another example.

 - Figure1: This figure is extremely useful but, the resolution and the quality are very low. Please increase the quality of this image.

 A: We appreciate reviewer’s suggestion. We have improved the image quality of Figure 1.

- Line 137: inhibiters? Maybe you can say “inhibitors”?

A: We appreciate reviewer’s remind. “Inhibiters” is typo and we have corrected it.

 - Line 148: …..NK cell regulatory factor, viz.,. Is it a typo?

 A: Viz. is the abbreviation of “videlicet”. We have rewritten this sentence.

- Line 151-155: This sentence is too large and hard to follow. Please, rephrase

A: We thank reviewer’s suggestion. The sentence has been shortened and re-organized at Line 152-155.

Reviewer 2 Report

The manuscript deals with a very interesting topic, but the approach should be improved in order to ameliorate the comprehension of all the information described in this paper.

In general, every section provides the information in a single paragraph, which makes its reading and comprehension very difficult. I suggest to the authors to introduce internal subsections or to divide the information in different paragraphs.

Also, I suggest to introduce one or two more figures showing a scheme summarizing the information provided in the text. For example, it should be interesting to add a figure showing the interactions between PCa cells and osteocytes.

The manuscript introduces two interesting points about:

  1. a) Which molecules or cells are key factors in choosing the bone metastasis type? The authors should remark and clearly summarize this point in the conclusions section.
  2. b) The therapeutic opportunities derived of a better knowledge of the pathways involved in bone metastasis development. The authors should also remark this point more widely in the conclusions section. In my opinion, the consequences of the hypothesis provided by Byrne et al (reference 87) indicating that castration causes bone loss and increased metastasis is very suggestive. The authors should indicate an explanation of the consequences related to these data.

Author Response

The manuscript deals with a very interesting topic, but the approach should be improved in order to ameliorate the comprehension of all the information described in this paper.

In general, every section provides the information in a single paragraph, which makes its reading and comprehension very difficult. I suggest to the authors to introduce internal subsections or to divide the information in different paragraphs.

A: We sincerely appreciate reviewer’s suggestion. We have divided each subsection into three-level subsection.

Also, I suggest to introduce one or two more figures showing a scheme summarizing the information provided in the text. For example, it should be interesting to add a figure showing the interactions between PCa cells and osteocytes.

A: We thank reviewer’s suggestion. In fact, we tried to draw a scheme to present dynamics of PCa/osteocyte interaction but it’s too complex to show in 2D diagram.

The manuscript introduces two interesting points about:

1. Which molecules or cells are key factors in choosing the bone metastasis type? The authors should remark and clearly summarize this point in the conclusions section.

A: We sincerely thank reviewer’s suggestion. We have added related description at Line 333-337. We think that it’s hard to judge that Ras or Wnt, which is the key factors of bone metastasis. But in bone homing stage, Wnt is more important than Ras signaling which Wnt lead osteoblastogenesis.

2. The therapeutic opportunities derived of a better knowledge of the pathways involved in bone metastasis development. The authors should also remark this point more widely in the conclusions section. In my opinion, the consequences of the hypothesis provided by Byrne et al (reference 87) indicating that castration causes bone loss and increased metastasis is very suggestive. The authors should indicate an explanation of the consequences related to these data.

A: We sincerely appreciate reviewer’s suggestion. We have added related description at Line 345-350.